# Human access impacts biodiversity of microscopic animals in sandy beaches

Alejandro Martínez [1,10], Ester M. Eckert[1,10], Tom Artois[2], Giovanni Careddu[3], Marco Casu[4], Marco Curini-Galletti[4], Vittorio Gazale[3], Stefan Gobert[2], Viatcheslav N. Ivanenko [5], Ulf Jondelius[6], Marinella Marzano[7], Graziano Pesole [7,8], Aldo Zanello[3], M. Antonio Todaro[9] & Diego Fontaneto [1✉]

Whereas most work to understand impacts of humans on biodiversity on coastal areas has focused on large, conspicuous organisms, we highlight effects of tourist access on the diversity of microscopic marine animals (meiofauna). We used a DNA metabarcoding approach with an iterative and phylogeny-based approach for the taxonomic assignment of meiofauna and relate diversity patterns to the numbers of tourists accessing sandy beaches on an otherwise un-impacted island National Park. Tourist frequentation, independently of differences in sediment granulometry, beach length, and other potential confounding factors, affected meiofaunal diversity in the shallow "swash" zone right at the mean water mark; the impacts declined with water depth (up to 2 m). The indicated negative effect on meiofauna may have a consequence on all the biota including the higher trophic levels. Thus, we claim that it is important to consider restricting access to beaches in touristic areas, in order to preserve biodiversity.

[1] Molecular Ecology Group (MEG), Water Research Institute (IRSA), National Research Council of Italy (CNR), Verbania, Italy. [2] Centre for Environmental Sciences, Hasselt University, Diepenbeek, Belgium. [3] Parco Nazionale dell'Asinara, Area Marina Protetta, Porto Torres, Italy. [4] Dipartimento di Medicina Veterinaria, Università di Sassari, Sassari, Italy. [5] Department of Invertebrate Zoology, Biological Faculty, Lomonosov Moscow State University, Moscow, Russia. [6] Department of Zoology, Swedish Museum of Natural History, Stockholm, Sweden. [7] Institute of Biomembranes, Bioenergetics and Molecular Biotechnologies (IBIOM), National Research Council of Italy (CNR), Bari, Italy. [8] Dipartimento di Bioscienze, Biotecnologie e Biofarmaceutica, Università degli Studi di Bari "A. Moro", Bari, Italy. [9] Dipartimento di Scienze della Vita, Università di Modena e Reggio Emilia, Modena, Italy. [10] These authors contributed equally: Alejandro Martínez, Ester M. Eckert. ✉email: diego.fontaneto@cnr.it

   1

Coastal areas and sandy beaches in particular are popular locations for recreational activities and holiday destinations, and are therefore subjected to intense stressors as a result of increasing urbanization and coastal infrastructure[1]. Marine protected areas have been created as a tool to protect biodiversity of marine and coastal environments from these alterations[2]. The creation of marine protected areas aims also at developing sustainable eco-tourism, which should have a positive effect on biodiversity and landscape[3], maintaining ecosystem services. Even if direct alterations of the ecosystem are reduced in such projects, tourism might nevertheless impact marine biota. Activities such as fishing or diving have conspicuous effects on the animal communities and have been widely investigated and regulated by environmental authorities[4]. Other apparently harmless human activities, such as the simple presence of people, could already affect animal communities, e.g. through the physical effect of trampling on the sand[5–9], the microbiological interference of human-related bacteria discharged in water[10,11], and the chemical release of pollutants from sunscreen creams[12].

Most of the studies addressing the impact of human presence on beaches were performed in highly urbanized and tourist-rich areas[13,14], where other stressors inherent to coastal development are present (e.g. pollution, coastal infrastructure, beach nourishment) potentially masking the effect of human presence on beach communities. Moreover, they were performed on large invertebrates, which often present hard shells and cuticles and can dig into the sediment layers beyond the one that is directly affected by tourists. In contrast, microscopic animals, collectively called meiofauna[15], are rich in species consisting of members of almost all animal phyla with a wide range of ecological features, differentially responding to human stresses, with a short generation time, allowing for a rapid detection of different types of impacts at different time scales[16–18]. Although these features make meiofauna a candidate to test the impact of human-driven changes in marine coastal areas[17], very few studies on the effect of tourists on meiofauna of sandy beaches are available[5], mostly because it is difficult to obtain reliable estimates of diversity on meiofauna dealing with morphological approaches in species identification[19]. The main problems are due to the vast diversity of meiofaunal organisms, with different extraction methods for each group to obtain animals for morphological identification[18], and to the lack of taxonomic expertise for several taxa making it almost impossible to identify meiofauna in biodiversity inventories[20].

We here investigate the effect of tourist presence on sandy beach ecosystems to provide quantitative estimates with the final goal of helping biological conservation of coastal areas, especially in countries that are highly impacted by tourism. Two problems have to be bypassed: on the one hand the presence of potential confounding factors other than human frequentation in the analysed area (e.g. pollution, coastal infrastructure, beach nourishment), and on the other hand the difficulties in using and identifying meiofauna. To minimize the effect of confounding factors, we selected the Asinara National Park in Sardinia, Italy as a study area, one of the least impacted localities with sandy beaches in the North-Western Mediterranean Sea. The area has no local inhabitants and has controlled access of tourists, present mostly during the summer season; far from cities and large harbours, the only human-induced impact on the beaches of the island is therefore due to tourists; moreover, activities other than walking on the beach and swimming in water, e.g. access to motorized vehicles, camping, beach grooming, are forbidden in the Park. To apply meiofauna as a metric for biological diversity, we used a DNA metabarcoding approach[19,21–24] from high-throughput sequencing of the V1–V2 region of the 18S rDNA with an iterative and phylogenetically informed taxonomic

assignment for the identification of the different taxa, paired with the identification of taxa also with a morphological approach for some selected groups of meiofauna.

Our hypothesis is that any effect of human presence on meiofauna living in sandy beaches will be detected more intensely where tourists have higher densities and can walk on the sand, namely in sand at the waterline in the swash level in the beach face (0 m depth), or in sand in shallow waters just below the low-tide shoreline (0.3 m depth), compared to sand in deeper waters (2 m depth), where people cannot walk but only swim.

## Results

**ZOTU and species diversity.** All 11 beaches of Asinara longer than 10 m (Fig. 1a, b) were sampled at three different depths (Supplementary Data 1), here called swash (on the beach face, at 0 cm water depth), shallow (below the low-tide shoreline, at 30 cm water depth), and deep (in the open water, at 2 m water depth, within a few metres offshore). A total of more than 460,000 high-quality merged reads clustered into 1069 ZOTUs (zero-radius operational taxonomic units); of these, 416 belonged to unicellular eukaryotes, 13 to non-meiofaunal metazoans (e.g. Asciacea, Porifera, Phoronida), and 640 to meiofauna (Supplementary Table 1). The 640 meiofaunal ZOTUs accounted for 60% of the ZOTUs and for 99% of the reads: almost all the reads in the dataset were indeed from meiofauna. Meiofaunal ZOTUs, assigned to taxonomic groups with an iterative approach, corresponded mostly to Nematoda (32.2% of the meiofaunal ZOTUs) and Copepoda (19.7%), followed by interstitial Annelida (12.3%), free-living Platyhelminthes (10.3%), Acoela (8.6%), and Gastrotricha (6.8%). These six groups (Fig. 1), representing 90% of the meiofaunal ZOTUs, were used also separately as major groups in subsequent analyses. The other groups, Acari, Gnathostomulida, Mollusca, Nemertea, Ostracoda, Rotifera, and Tardigrada were represented by 3–14 ZOTUs, Kinorhyncha by a single ZOTU (Supplementary Table 1).

**Richness.** ZOTU richness per sample ranged from 24 to 180 (Supplementary Table 1). The overall number of meiofaunal metazoan ZOTUs was significantly different at different depths (generalized linear mixed effect model, GLMEM; depth: LR chi-squared = 25.4, $p < 0.0001$), with more ZOTUs in sand in deep water (Fig. 2), and at different beaches (likelihood ratio chi-squared, LR = 31.4, $p < 0.0001$), without being affected by the potential bias due to number of reads per sample (LR = 0.8, $p = 0.380$). Analysing each major meiofaunal group separately, results were congruent with the overall analysis on all ZOTUs, except for Gastrotricha and Copepoda, for which richness was not affected by differences between beaches, and Acoela, whose richness was not affected by depth and was marginally related to the number of reads (Supplementary Table 2), suggesting a complex scenario of potential drivers of ZOTU richness in different beaches at different water depths for the different taxa.

Thus, to be able to understand the actual patterns of diversity, we build a second set of models in which we analysed the effect of tourists against the total number of ZOTUs separately for each depth level (swash, shallow, and deep), including the length of the beach and the type of sediment granulometry as confounding factors. We found a negative relationship between the total number of ZOTUs and the number of tourists at swash and shallow levels, although the relationship was significant only at the swash level (generalized linear model, GLM: LR = 4.8, $p = 0.028$) (Table 1, Fig. 2). Significant negative correlations were found between number of ZOTUs and number of tourists for Copepoda at the swash (LR = 4.6, $p = 0.031$) and for Acoela at the shallow level (LR = 5.9, $p = 0.015$), whereas no clear

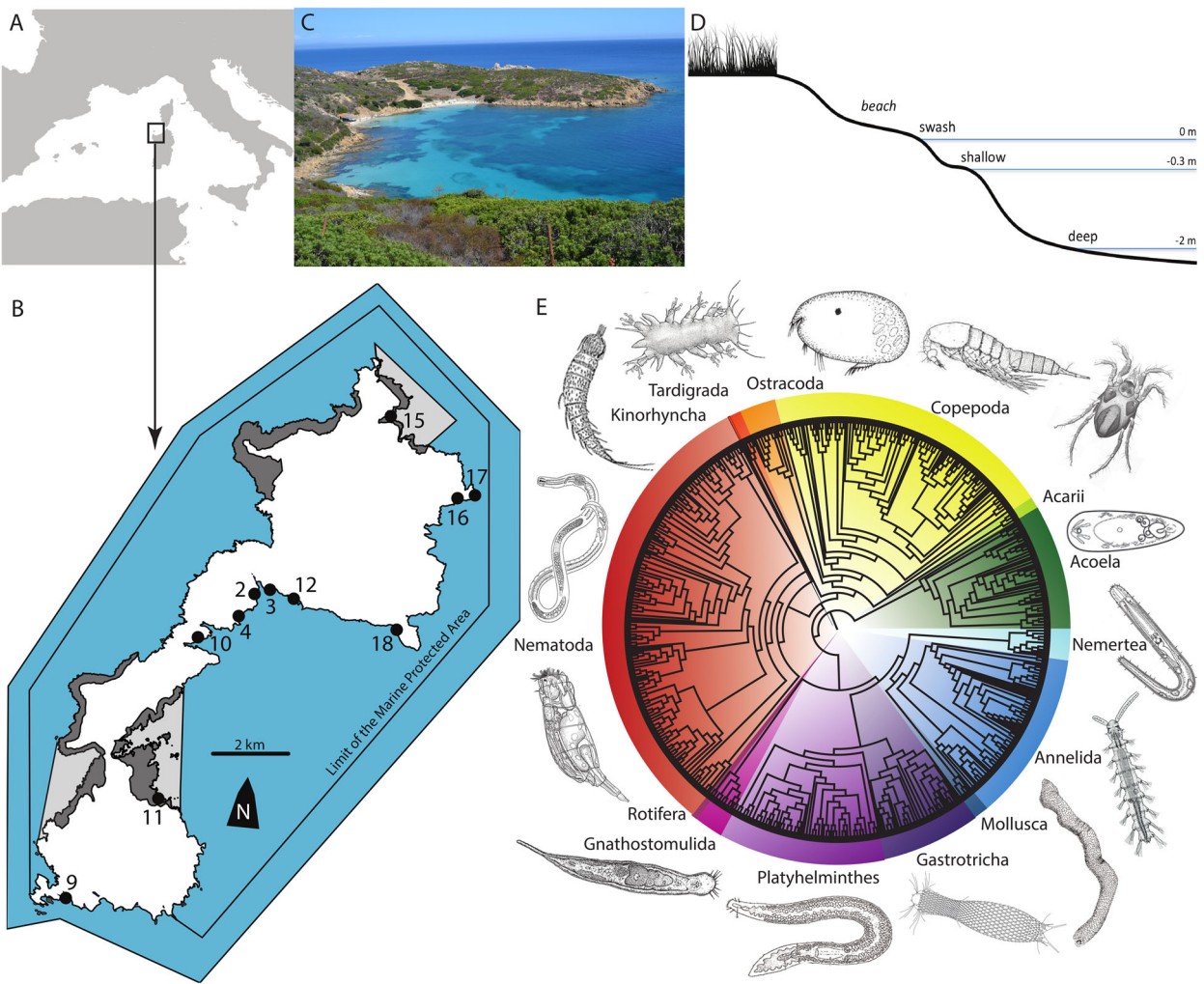

**Fig. 1 The Asinara National park and its marine meiofauna. a** Map of the Western Mediterranean showing the Asinara National Park. **b** Map of Asinara Island showing the limits of the Marine Protected Area. Grey areas represent areas with restricted access for tourists: dark grey for terrestrial habitats and light grey for marine habitats. Sampling locations are coded as in Supplementary Data 1. **c** Cala dei Ponzesi (beach 16), a beach open to tourists and with high presence of people. **d** Profile of the sampling scheme for each beach. **e** Neighbour joining phylogenetic reconstruction of all 640 meiofauna ZOTUs (zero-radius operational taxonomic units) found in the sandy beaches of Asinara, divided by the 14 taxonomic groups. Numbers on the map in **b** correspond to the following sampled localities: 2, Cala Stagno Lungo West; 3, Cala Stagno Lungo East; 4, Muro Lungo; 9, Cala Spalmadori; 10, Cala Marcutza; 11, Cala Sant'Andrea; 12, Cala between Cala Tonda and Cala Reale; 15, Cala d'Arena; 16, Cala dei Ponzesi; 17, Cala Giordano; 18, Cala Trabuccato.

information was found for the other taxa (Supplementary Table 3).

In contrast, the analyses using the morphological dataset at each level did not show clear results, with no significant relationships recovered between the total number of species and the number of tourists at any water depth (Supplementary Table 4). Only the richness of Annelida was positively correlated with number of tourists at the swash level.

**Community composition**. Mirroring the results obtained for the meiofaunal richness analyses, our analyses also suggest complex scenario of potential drivers of the meiofaunal community composition at different beaches with different number of tourists interacting at different depths. As expected, changes in the overall ZOTU composition of the meiofaunal communities, analysed with presence/absence data, were better explained by the inherent differences between beaches (PERMANOVA: $R^2 = 0.381$, $p = 0.0038$), followed by differences in sediment granulometry ($R^2 = 0.263$, $p = 0.002$). We detected a significant effect of the number

of tourists, even if it explained a smaller proportion of the variability in community composition ($R^2 = 0.058$, $p = 0.0021$) (Table 2).

A second set of models testing the effect of the number of tourists at each level and including sediment granulometry as a confounding factor revealed that the explanatory power of number of tourists was higher in the swash ($R^2 = 0.184$, $p = 0.204$) and the shallow ($R^2 = 0.171$, $p = 0.241$) levels than in the deep one ($R^2 = 0.111$, $p = 0.204$), even if not statistically significant. Such scenario was consistent across all meiofaunal groups (Supplementary Tables 5 and 6).

**Phylogenetic diversity**. No clear effect of the explanatory variables was found on phylogenetic diversity and mean phylogenetic distance, except for the number of ZOTUs significantly affecting phylogenetic diversity of all the groups, and sediment granulometry affecting phylogenetic diversity of Annelida (GLM: LR = 36.1, $p < 0.0001$) and Nematoda (LR = 19.8, $p < 0.0001$) (Supplementary Table 7).

## Discussion

Our results suggest that the presence of tourist on beaches may have an impact on the communities of microscopic animals: a negative correlation between the number of meiofaunal ZOTUs and the number of tourists on beaches could be seen for the swash (slightly significant) and the shallow level (even if not significant). We thus found a stronger effect of tourists at the swash level, but we cannot disentangle whether this could be due only to a higher presence of tourists in the swash level than in the other two water levels or if this habitat is indeed more sensitive to human presence. Changes in the taxonomic composition of meiofauna, as expected from previous studies[15,17–19], were due mostly to differences between beaches and sediment granulometry, but also to the presence of tourists and not so much to the differences between depths. We suggest that the effect of presence of people could be assigned mostly to trampling, since walking on the beach at the swash level is the major

disturbance activity of humans in the sampled areas, where other more impacting recreational activities (e.g. access to motorized vehicles, camping, beach grooming, etc.) are forbidden. Other potential effect could be indirectly related to human presence, such as to the amount of sunscreen cream and of faecal-related bacteria that enter the water, which are likely proportional to the number of tourists. We acknowledge that sample size for the most densely frequented beaches in Asinara is low (Fig. 2); yet, the statistical approach is robust and accounts for potential confounding factors. In any case, the negative effect of tourists was always present, more significantly at the swash level, where people indeed walk on the beach, and only marginally present in shallow waters but completely absent in deep waters. Such effect was visible even for the low density of tourists that are present in the near-pristine habitats of the Asinara National Park. A maximum of 300 people per day walking every $10\,m^2$ means that during all the day less than 30

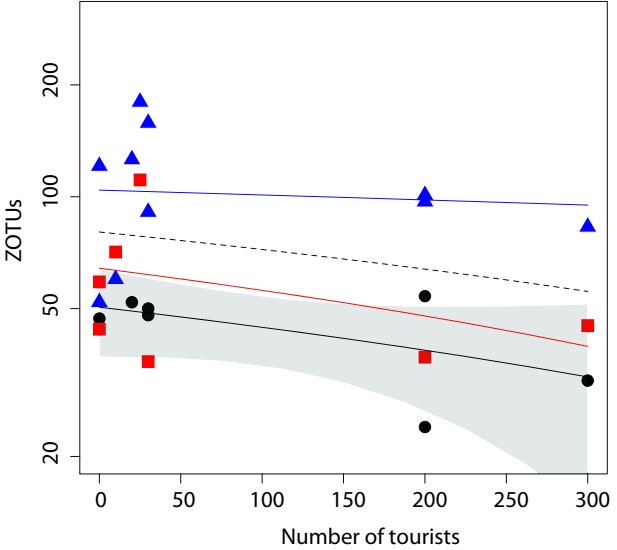

**Fig. 2 Number of meiofaunal ZOTUs (on a logarithmic scale) in relationship to the maximum number of tourists per day every 10 m².** The black dashed line represents the (non-significant) regression line for the entire dataset. Points and lines represent the relationship for each water depth: swash (black dots), shallow (red squares), and deep (blue triangles). The significant regression line for swash water level is reported with 95% confidence interval.

**Table 2 Effect of tourists and other variables on meiofaunal ZOTU community composition.**

| Predictor | $R^2$ | P |
|---|---|---|
| *All water depths* | | |
| Tourists | 0.058 | **0.021** |
| Granulometry | 0.263 | **0.002** |
| Depth | 0.090 | **0.003** |
| Beach | 0.381 | **0.004** |
| Residual | 0.202 | |
| *Swash* | | |
| Tourists | 0.184 | 0.204 |
| Granulometry | NA | NA |
| Residual | 0.816 | |
| *Shallow* | | |
| Tourists | 0.171 | 0.241 |
| Granulometry | 0.678 | 0.200 |
| Residual | 0.151 | |
| *Deep* | | |
| Tourists | 0.111 | 0.204 |
| Granulometry | 0.404 | 0.366 |
| Residual | 0.486 | |

Results are reported from permutational multivariate analyses of variance (PERMANOVA) on the effect of different sets of explanatory variables on community composition calculated as pairwise Jaccard dissimilarities of the ZOTU occurrence. Analyses are performed for all the dataset, as well as the swash, shallow, and deep level. Differences in granulometry were not analysed at the swash level because only one granulometry type was present for the level. $R^2$ and *p* values are reported. *P* values for significant predictors are marked in bold.

**Table 1 Effect of the number of tourists together with the potential confounding factor of type of sediment granulometry and beach length on the richness of meiofaunal ZOTUs (zero-radius operational taxonomic units) at the three water depths, according to a type II ANOVA output of generalized linear models.**

| Predictor | LR Chisq | Estimate ± s.e. | df | P |
|---|---|---|---|---|
| *Swash* | | | | |
| Tourists | 4.786 | −0.170 ± 0.078 | 1 | **0.028** |
| Granulometry | NA | NA | NA | NA |
| Length | 0.003 | −0.004 ± 0.072 | 1 | 0.956 |
| *Shallow* | | | | |
| Tourists | 0.000 | −0.005 ± 0.300 | 1 | 0.987 |
| Granulometry | 53.428 | NA | 4 | **<0.0001** |
| Length | 0.003 | −0.017 ± 0.298 | 1 | 0.957 |
| *Deep* | | | | |
| Tourists | 2.388 | 0.070 ± 0.045 | 1 | 0.122 |
| Granulometry | 101.570 | NA | 4 | **<0.0001** |
| Length | 0.060 | 0.011 ± 0.043 | 1 | 0.807 |

*LR Chisq* likelihood ratio chi-square values, *df* degrees of freedom, *P* chi-square goodness of fit, *s.e.* standard error. Differences in granulometry were not analysed at the swash level because only one granulometry type was present for the level. *P* values for significant predictors are marked in bold.

people passed on every square metre at the peak of the tourist season.

Previous studies on meiofauna or on larger invertebrates were not always consistent in their results, and provided evidence of human disturbance only at very high levels of frequentation by tourists[6–9]. Moreover, they were performed in urbanized areas where other stressors continuously affect beach ecosystems throughout the year potentially masking the effect of people during the touristic season[5,8]. This is not the case in the Asinara National Park, never populated or subjected to intensive touristic exploitation. The use of the metabarcoding approach with an iterative and phylogeny-informed taxonomic assignment allowed us to overcome the impediment to describe meiofaunal communities using morphological approaches, in contrast with other studies that had to focus on certain groups only, often copepods or nematodes, which are easier to extract and to preserve[15], neglecting the most sensitive soft-bodied taxa[5], e.g. flatworms, gastrotrichs, acoels, gnathostomulids, rotifers, for which sampling and preservation for morphological analyses is more problematic[18]. Metabarcoding from meiofauna is now a common approach in biodiversity studies[21,25] and our results demonstrate that we can strengthen the support for the use of metabarcoding of meiofauna in routine environmental monitoring, potentially not only for sandy beaches[26,27]. We found an indication of the effect of human presence notwithstanding the use of one single marker and with a certain proportion of samples that did not work, potentially because of non-optimal processing and storage before DNA extraction. Using appropriate storage and a multi-marker approach[28,29] may work even better. The metabarcoding protocol can be considered more efficient than the approach based on morphological identification of species: especially for meiofauna, metabarcoding is faster, less subjective, and mostly cheaper than a morphological approach[25]. We managed to handle all meiofaunal groups for all beaches for metabarcoding, whereas the same team of taxonomic experts who worked for this study could handle only few groups and for only a selection of the beaches (Supplementary Data 2). As a negative side, the problem of metabarcoding of meiofauna is that no abundance data can be considered reliable, and because of occurrence-only data, biological monitoring through metabarcoding is still under discussion for freshwater and marine habitats[30], but will surely become a reality in the near future[31]. In our case, with at least 88 of the 196 morphologically identified species being potentially new for science (Supplementary Data 2), the process of morphological identification for biological monitoring is massively slowed down. Due to such high number of yet unknown species with no DNA sequence data, the taxonomic assignment of metabarcoding data may be flawed. What is known from previous studies is that metabarcoding and morphology may provide different results for the assessment of biological diversity[19,21,23]. We confirm such indication, showing that several genera and families have been found uniquely with metabarcoding or with a morphological approach for the six taxonomic groups on which both approaches were applied (Fig. 3). In addition, it is known that the 18S rDNA marker we selected may underestimate diversity for some taxa[32], potentially explaining the low number of ZOTUs for some groups such as Rotifera and Kinorhyncha. Regardless of potential problems and differences between different approaches, one of the main messages of our study is that we confirm the use of metabarcoding on meiofauna as a tool for biodiversity monitoring[19,21,22,25].

The other main message of our results is that if human access may impact biodiversity beaches with restricted access to tourists should be considered when planning marine reserves, in order to preserve biodiversity, especially in an impacted but highly species-diverse area such as the Mediterranean Sea[33]. Since many meiofaunal taxa are restricted to sandy beaches and may be sensitive to trampling and other indirect influences from the mere

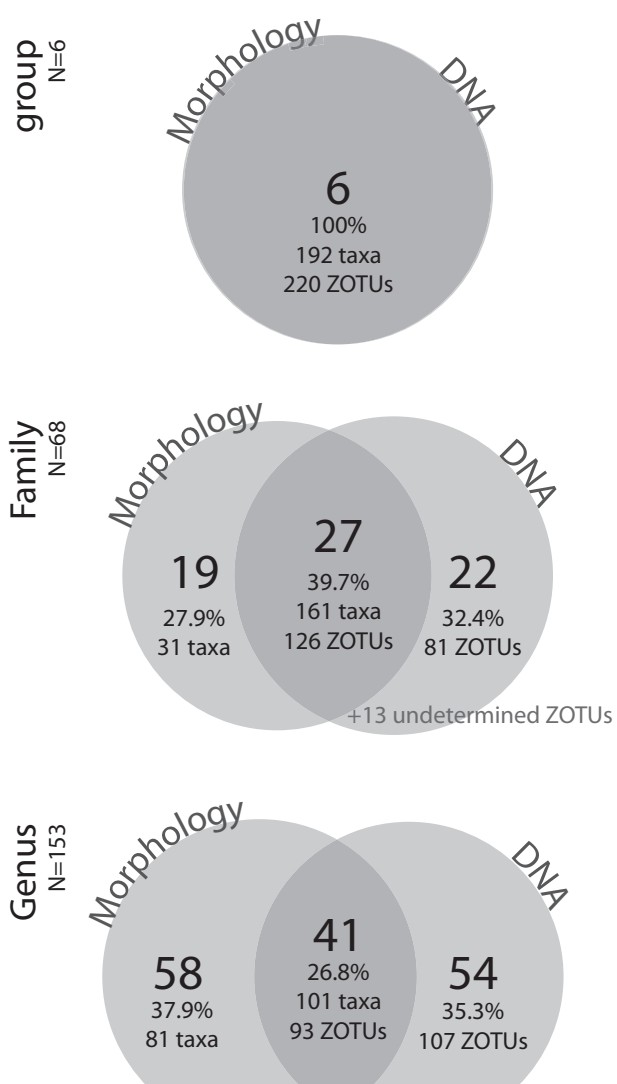

**Fig. 3 Venn diagrams at the group, family, and genus level.** Venn diagrams at the group, family, and genus level, to compare how much of the diversity was shared between a traditional morphological approach in identifying taxa and a DNA-based metabarcoding approach to identify ZOTUs, limited to the six main groups (Acoela, Annelida, Gastrotricha, Nemertodermatida, Proseriata, Rhabdocoela) that we analysed both on morphology and on metabarcoding. At the largest taxonomic level, all six groups were found from morphology (192 taxa) and from DNA metabarcoding (220 ZOTUs); at the family level, slightly less than 40% of the total 68 identified families was found by both methods; at the genus level, less than 30% of the total 153 identified genera was found by both methods. Note that a certain number of taxa and ZOTUs could not be assigned to family or genus level, but only to the larger groups. The number of taxa and ZOTUs is also reported.

presence of people, our results highlight the necessity of implementing management strategies including integral protection for specific sandy beaches for conservation purposes. Protection can consist either of including certain beaches into areas of integral protection within marine protected areas, as is the case in Asinara, or alternatively by defining specific zones within beaches in which tourist access is forbidden. This last approach has already been implemented in areas with extensive coastal sandy areas with positive effects on coastal marine communities[14]. However,

in areas in which most sandy beaches are short, such as the Mediterranean, efficient subdivision of beaches might not be possible. In such areas, integral protection within marine protected areas seems the only solution, but it needs to be carefully planned and supported by campaigns of social awareness on the importance of such a measures[34], potentially using metabarcoding as a tool to show the effect of human presence.

From an ecological perspective, the negative effects on meiofauna that we highlighted may not be strictly limited to these microscopic animals: any negative impact on them will have a general effect on the ecosystem functioning[35,36]. Meiofaunal organisms play a key role in production, consumption, decomposition, nutrient regeneration, and energy transfer, but also in preventing sediment erosion in sandy beaches[37]. It is true that we highlighted a negative effect on meiofauna only at the swash zone, thus, not allowing any supported inference on the effects on ecosystem functioning of the beach ecosystem.

In conclusion our results have two important implications: (1) The use of biological monitoring through DNA metabarcoding in aquatic habitats should be pursued as an efficient and reliable future possible methodology to identify ecosystem pressures[31] and inform environmental stakeholders and politicians in their decision-making process. (2) Even low intensity of presence of tourists might produce an overall loss of biodiversity, especially in the sensitive groups of microscopic animals; it is possible that such effect could not be demonstrated any more in heavily disturbed coastal areas, where most beaches are frequently trampled and the most sensitive organisms would have disappeared already, but our sampling in Asinara allowed us to identify such effects, even if we acknowledge that the results are only slightly significant and limited to the swash zone.

## Methods

**Sampling design**. The Asinara National Park (http://www.parks.it/parco.nazionale.asinara/Eindex.php), located at the island of Asinara in the North-Western tip of Sardinia, Italy, covers a Marine Protected Area of about 110 km$^2$. Before a National Park was established in 1997, the island hosted a hospital in the nineteenth century and then a prisoner camp and a maximum-security prison from 1885 to 1997 (ref. [38]); nobody lives permanently on the island, and the only people who can impact marine biodiversity are tourists. The National Park receives an average of less than 1000 tourists every day during summer and almost no tourists from October to March; the total number of tourists is anyway limited, providing one of the best examples of potentially sustainable tourism along the coastline of the Mediterranean Sea[3,38]. Meiofauna on the island, even if disturbed by tourists, may have the possibility to recover during the tourist-free period from October to May. Asinara island is sinus shaped with four mountainous sections linked by a narrow, flat coastal belt (Fig. 1b). The tidal range in the area is just a few centimetres, making the physical impact of tides on meiofauna very limited. The west side of the island is rocky and steep, while the east side has flat areas occupied by coves and beaches. Most of the 11 beaches longer than 10 m are open to tourists, except for the two within the areas of total protection, corresponding to "Cala Sant'Andrea e Cala di Scombro di Dentro" at the centre and "Cala Arena e Punta dello Scorno" at the northern tip of the island (Fig. 1b).

We sampled all 11 beaches longer than 10 m present in the park (Fig. 1b). The beaches are mostly pocket beaches, between 20 m and 400 m along the shoreline, relatively homogeneous in their physical, ecological, and geographic conditions (Fig. 1d): they are within a maximum distance of 15.5 km, which minimizes spatial and biogeographic confounding factors; they are on the more protected Eastern coast of the island, which minimizes ecological factors of physical exposition to waves; and they are all sandy beaches, which minimizes ecological differences due to sediment granulometry. The major difference between the beaches is the number of tourists received during the summer months (Supplementary Data 1). The daily affluence of tourists ranged from beaches with no tourists to beaches with peaks of 300 tourists per day every 10 m$^2$, even if such relatively high numbers were reached for only one or a few days in the season. Two major beaches (Cala Sant'Andrea and Cala d'Arena) within the areas of integral protection are restricted to the public (Supplementary Data 1). The number of tourists per beach was estimated by the authorities of the park by direct observations and the data are stored in their unpublished archives. The samples for the extraction of meiofauna were collected at the end of the tourist season between 22 September and 1 October 2014.

Sediment samples were collected manually. Each sample consisted of four replicates of 1 liter of sediments from the upper 5 cm of sand collected over a homogenous area of 1 m$^2$ by scooping the top layer of sand with a jar. Immediately

after collection, samples were taken to the laboratory on the island. All samples were processed within few hours after collection. Total meiofauna for metabarcoding with high-throughput sequencing (SI Appendix) was extracted from two replicates using the MgCl$_2$ decantation technique[15] through a mesh size of 63 μm and immediately preserved in ethanol at −20 °C. The other two replicates were used one for the analysis of sediment granulometry and one for morphological identification of meiofauna (see below).

**Metabarcoding**. Overall, 11 beaches were samples, with three levels (here called swash, shallow, and deep). Each of the 33 samples was sequenced twice for a total of 66 sequencing reactions; then, the replicate with the highest number of reads for each of the 33 samples was used for metabarcoding. Eight of the total 33 samples were discarded (Supplementary Data 1), due to the low quality of the DNA in both replicates. Sequence reads are publicly accessible at NCBI (GenBank) with accession number PRJNA369046. Index, adapter, and primers were removed with cutadapt 1.9.1 (ref. [39]). The UPARSE pipeline was used for merging of sequences and quality control; USEARCH for the clustering of operational taxonomic units (OTUs)[40,41]. The pipeline was essentially used following the author's online tutorial with the following settings: when merging sequences the maximum number of nucleotides that were allowed to be different in the overlap (max-diff) was set to 10 and the merged sequences had to have a minimum length of 300 bp. ZOTUs were calculated using the UNOISE algorithm, which attempts to identify all correct biological sequences in the reads (high-quality requirements and more than eight reads in the dataset) and cluster the other sequences around them, resulting in presumed 100% sequence identity termed ZOTUs for *zero-radius OTU*[42,43]. No rarefaction of the original raw data was performed in order to maintain all the sequences we obtained[44]; yet, we explicitly tested for the potential confounding effect of number of reads in the statistical tests (see below).

We used an iterative and phylogenetically informed approach for the taxonomic assignment of ZOTUs. Non-metazoan meiofaunal sequences, identified through Blast against the whole GenBank database, were discarded; meiofaunal metazoan sequences were assigned to the following 14 major taxonomic groups: Acari, Acoela, Annelida, Copepoda, Gastrotricha, Gnathostomulida, Kinorhyncha, Mollusca, Nematoda, Nemertea, Ostracoda, Platyhelminthes, Rotifera, and Tardigrada. Some of them, namely Acoela, Annelida, Gastrotricha, Gnathostomulida, Mollusca, Nemertea, Platyhelminthes, and Rotifera, are considered soft-bodied, are difficult to sample and preserve, and are often neglected in meiofaunal studies based on morphology[18]. Sequences were posteriorly assigned to each group using a neighbour joining tree including all recovered ZOTUs (Supplementary Fig. 1), aligned using a Q-ins-i refinement method implemented in MAFFT version 7 (ref. [45]). Blast taxonomic assignments were then confirmed by building separated neighbour joining trees for each taxonomic group, adding all the overlapping sequences available in GenBank for each group at the date of March 2019 (Supplementary Figs. 2–20). Such additional analyses allowed us to be more confident about the taxonomic assignment of each ZOTU to each group at the desired taxonomic level (phylum, class, or order), regardless of the species or genus assignment, due to the potentially high number of unknown species in meiofauna[18]. Sequences were downloaded from GenBank, added to our dataset and handled for the analyses using the R packages rentrez 0.4.1 (ref. [46]) and ape 3.2 (ref. [47]). Only the ZOTUs that were eventually unambiguously nested within their target groups including identified GenBank sequences were retained, corresponding to the 99.9% of the metazoan sequences, and then the ZOTUs of non-strictly meiofaunal groups, e.g. Cnidaria, were removed (Supplementary Table 1).

Phylogenetic diversity was assessed from ultrametric trees obtained by BEAST package v2.4.8 (ref. [48]) (Supplementary Figs. 21–26), using a Yule Process for tree priors and a generalized time-reversible (GTR) model for nucleotide evolution including a gamma distributed rate of variation among sites. Four chains run for 50 million generations, sampled every 10,000. Consensus trees were obtained after confirming convergence and discarding 10 million trees as burnin.

**Morphological analyses**. One of the four replicates collected for each sample was used to perform a parallel analysis on the effect of human frequentation on species identified using morphological criteria. These samples were processed mostly using the MgCl$_2$ decantation technique but also by siphoning off the water just above the sediment, and using small variations in the methods, according to the focal taxon of study, as in previous studies covering different meiofaunal groups from the same samples[18]. Live material was studied using dissecting and compound light microscopes. Additional material for identification and/or descriptive purposes was preserved using methods appropriate for the respective taxon.

Due to the constraints in available taxonomic expertise and the long time that is needed to identify the whole species assemblages for each taxon, we focused on four main meiofaunal groups (Acoela, interstitial Annelida, Gastrotricha, and Platyhelminthes) only on six beaches (Supplementary Data 2). Identification of the specimens was performed using taxonomic keys and original literature according to the state-of-the-art systematics of each group.

**Explanatory variables**. As a proxy to account for the effect of human frequentation we used the maximum number of tourists for each 10 m$^2$ of the analysed beaches, as measured by the records of the surveillance personnel of the park. Water depth was considered as a categorical explanatory variable (three fixed levels:

0 m, called swash; 0.3 m, called shallow; 2 m, called deep). Other potentially confounding factors that we included in our statistical models were: number of reads, intrinsic differences between beaches, and interactions between these factors, in addition to beach length and differences in sediment granulometry. Granulometry was assessed by passing 150 g of dry sediment through six sieves with mesh sizes corresponding to a range from 1 mm to 50 μm, shaking, fractioning, and weighing to obtain mean grain size, sorting coefficient, kurtosis, and skewness[49,50] (Supplementary Data 1). From such measurements, we grouped sediments by granulometry by a k-means analysis to find the optimal number of groups, selected using Bayesian Information Criterion (BIC) for expectation–maximization algorithm initialized by hierarchical clustering for parameterized Gaussian mixture models[51] using the mclust v5.3 R package[52]. All values were scaled before performing the analysis[53]. The highest BIC was obtained for 7 groups (BIC = −106.6, EEV ellipsoidal, equal volume, and equal shape multivariate mixture model). We accounted also for the effects of different length of each beach on the meiofaunal composition and richness by including such measure in the models.

**Response variables**. The effect of tourists was evaluated on three different types of community descriptors, included as response variables in the different sets of models: richness, community composition, and phylogenetic diversity. Richness was measured as the number of ZOTUs (or morphological species) for the total meiofauna and for each major group (defined as representing at least 5% of the total ZOTUs). Community differences between samples were measured using the Jaccard dissimilarity index from binary presence/absence data calculated with the R package betapart v. 1.5.1 (ref. [54]). Phylogenetic diversity was measured as diversity and sorting at the phylogenetic level measured on ultrametric BEAST trees calculated only for the six taxonomic groups with more ZOTUs, in order to avoid biases due to low taxonomic diversity in the phylogenies. Phylogenetic diversity was calculated as Faith's phylogenetic diversity (the sum of the total phylogenetic branch length for one or multiple samples) and as phylogenetic clustering (standardized effect size of the mean phylogenetic diversity (MPD), equivalent to 1-Nearest Relative Index, NRI[55]), with the R package picante 1.6-2 (ref. [56]).

**Statistical models**. We developed statistical models to test the effect of tourists on meiofauna richness, community composition, and phylogenetic diversity. In order to mirror the complex structure of biological reality, our models included additional explanatory variables that could affect the response variables. To be able to account for a combination of such accounted and unaccounted effects in the models with richness and phylogenetic diversity as a response variable, we used GLMEMs, designed exactly for these kinds of analyses, with violations of the assumption that data are independent[57]. Thus, in the first set of GLMEMs we used number of tourists per 10 m$^2$, depth (three levels: swash, shallow, deep), sediment granulometry (7 levels), and beach length, as explicit explanatory variables; the identity of the 11 beaches was included as a random effect to account for unmeasured differences between beaches and for spatial auto-correlation between samples within each beach. Then, we explored the effect of human presence separately for each depth using GLMs, including number of tourists per 10 m$^2$, length of the beach, and type of sediment granulometry as explanatory variables. Before performing such detailed analyses, we assessed: (1) whether the assumption of differences between beaches and water depths held true, or rather they could depend on the confounding factor of number of reads by analysing their effect on each response variables using GLMs; (2) whether explanatory variables were correlated (they were not: absolute $r$ values were below 0.52).

For GLMEMs and GLMs with ZOTU and morphological species richness as the response variable, we assumed a Poisson error structure in the models; for GLMs, we assumed a quasi-Poisson error structure when we found evidence of data overdispersion. A Gaussian error structure was implemented for all models with phylogenetic diversity and phylogenetic sorting as the response variable. Model fit was checked for GLMs by plotting model residuals; plotting the predicted versus fitted residuals; using the normal Q–Q plot; checking Cook's distances[53]. For GLMEMs, we checked the predicted versus fitted residuals. Results are presented always as Analysis of Deviance Tables from the R package car 2.1-3 (ref. [58]). Analyses of Deviance Tables give a clear message on the effect of each variable, calculating the significance using likelihood ratio (LR) chi-square tests for GLMs and Wald (W) chi-square tests for GLMEMs[58].

For models on community composition using matrices of Jaccard pairwise differences as a response variable, we assessed the percentage of the variability in community composition observed across samples using a permutational multivariate analysis of variance (PERMANOVA) in the R package vegan 2.2-1 (ref. [59]). The structure of the models for community composition followed the same rationale of the analyses on ZOTU richness. All analyses were performed in R 3.6.3 (ref. [60]).

**DNA extraction and sequencing**. DNA was extracted for metabarcoding from all 11 beaches. Each sample was vortexed for 10 s after which 6 ml were transferred to a small Petri dish and the ethanol was evaporated on a 60 °C hot plate on a sterile bench (approx. 2–5 h). An aliquot of 0.5 ml of extraction buffer (0.1 M Tris-HCl, 0.1 M NaCl, 0.1 M EDTA, 1% SDS, and 250 μg/ml proteinase K) was added to the dry sample, which was then incubated for 2 h at 56 °C. The sample was then resuspended by vigorous pipetting and all the liquid was transferred to the micro-

bead tube of the commercial PowerSoil extraction kit (MoBio). The kit was used according to the manufacturer's protocol, except for the last step (elution of DNA from the spin column), for which twice 25 μl of elution buffer were incubated on the spin filter column for 15 min before centrifugation.

The primers used for Illumina sequencing were based on 18SF04 and 18SR22 (ref. [61]). The selected primers amplify a DNA fragment of approximately 450 base pairs corresponding to the V1–V2 regions of the nuclear small subunit rRNA gene (18S rDNA). The coverage of the primer was, however, verified using the ARB software package[62] with the SILVA reference database release 111 (ref. [63]) and an ambiguous base was added to the reverse primer 18SR22: 5′-GCCTGCTGCCTTC CTTRGA-3′.

The DNA extracted from each sample was used as a template for amplicon library preparation, adopting a modified version of the Illumina Nextera protocol[64]. In particular, the library preparation was based on two amplification steps. In the first amplification, V1–V2 regions were amplified using the universal primers, reported above, having a 5′ end overhang sequence, corresponding to the Nextera Transposon Sequences (Illumina Adapter Sequences Document, Document # 1000000002694 v01 February 2016). Amplifications were performed using the Phusion High-Fidelity DNA polymerase system (Thermo Fisher Scientific). Each reaction mixture contained 0.5 ng of extracted DNA, 1× Buffer HF, 0.2 mM dNTPs, 0.5 μM of each primer, and 1 U of Phusion High-Fidelity DNA polymerase in a final volume of 50 μl. The cycling parameters for PCR were standardized as follows: initial denaturation 98 °C for 30 s, followed by 12 cycles of 98 °C for 10 s, 50 °C for 30 s, 72 °C for 15 s, and subsequently 18 cycles of 98 °C for 10 s, 62 °C for 30 s, 72 °C for 15 s, with a final extension step of 7 min at 72 °C[65]. All PCRs were performed in triplicate and in the presence of a negative control (Molecular Biology Grade Water, RNase/DNase-free water). The PCR products were visualized on a 1.2% agarose gel and purified using the AMPure XP Beads (Agencourt Bioscience Corp., Beverly, MA, USA), at a concentration of 1.2× vol/vol, according to the manufacturer's instructions. The purified amplicons were used as templates in the second PCR round, which was performed with the Nextera indices priming sequences as required by the dual index approach reported in the Nextera DNA sample preparation guide (Illumina). The 50 μl reaction mixture was made up of the following reagents: template DNA (40 ng), 1× Buffer HF, dNTPs (0.1 mM), Nextera index primers (index 1 and 2), P5 and P7 primers at 0.2 μM and 1 U of Phusion DNA Polymerase. The cycling parameters were those suggested by the Illumina Nextera protocol. The dual indexed amplicons were purified using AMPure XP Beads, at a concentration of 0.8× vol/vol checked for quality control on a 2100 Bioanalyzer (Agilent Technologies, Santa Clara, CA, USA) and quantified by the fluorometric method using the Quant-iTTM PicoGreen-dsDNA Assay Kit (Thermo Fisher Scientific, Waltham, MA, USA) on a NanoDrop 3300 (Thermo Fisher Scientific). Metagenomic libraries obtained were normalized to the 2 nM concentration, pooled, and sequenced on the MiSeq Illumina platform using the 2 × 300 paired-end (PE) approach. In order to increase the genetic diversity, as required by the MiSeq platform, a 5% of the phage PhiX genomic DNA library and a 40% of other genomic DNA libraries were added to the mix and co-sequenced.

**Reporting summary**. Further information on research design is available in the Nature Research Reporting Summary linked to this article.

## Data availability
Sequence reads are publicly accessible at NCBI (GenBank) with accession number PRJNA369046. Any other data are reported in Supplementary Data Files and can also be requested from the authors with no restriction to access.

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

## Acknowledgements

This research has received funding from the IEF Marie Skłodowska-Curie Actions of the EU's Horizon2020 program, grant No. 655537—RAVE to E.M.E, grant No. 745530—ANCAVE to A.M. Sequencing was financed through the Laboratorio di Biodiversità Molecolare—Lifewatch Italy call. We particularly thank the personnel at Asinara National Park in Sardinia, Italy, for logistic support and sampling permissions.

## Author contributions

A.M., T.A., G.C., M.C., M.C.-G., V.G., S.G., U.J., A.Z., M.A.T., and D.F. collected data. A.M., E.M.E., T.A., G.C., M.C., M.C.-G., V.G., S.G., V.N.I., U.J., M.M., G.P., A.Z., M.A.T., and D.F. performed analyses. A.M., E.M.E., and D.F. wrote the first draft of the manuscript. All authors contributed to initial manuscript conception and final editing.

## Competing interests

The authors declare no competing interests.
