## [Peer Review File · Communications Biology]

Reviewers' comments:

Reviewer #1 (Remarks to the Author):

The study is analyzing the effect of the human impact of trampling, like a gradient of tourism on Meiofauna diversity at the beach. Analyzed are different beaches at an Island in Italy, with different restrictions of tourists by sampling the three different depth (0cm, 30cm, 2m). The methods are comprehensive including metabarcoding (18S), morphologically based analyses, as well as granulometry. The extensive statistical analyses showed negative indications especially for copepods and Acoela of tourists walking along the waterline. Even if the sampling size is not high and samples were not in optimal condition, the study shows that the metabarcoding approach is a valid and faster method for biodiversity detection of sandy beaches and can be included in the further environmental and touristic decision-making process.

Overall the study is convincing and important for that field. It is given a clear question, important for politics and economy, and up to date methods able to answer that. It is a nice example of how we can use Metabarcoding and how to correctly use them for statistical evaluations.

Within the study, a strong hypothesis is given, nice results and a clear conclusion. The study is valid, based on useful methods and data, but they are not presented in a way that it is obvious. The parts do not seem fitting to each other and the central theme got lost (especially in the introduction and methods). Therefore, I would suggest rewriting parts of the manuscript and being more specific, to better present the scientific context of this study. As I am not a native speaker I can and will not correct language or grammar, however, a style check is always recommended.

The text focused a lot about value tourism in the area, however, it seems repeating, suggestive and not on point. There are clear data for that but in the supplementary and not clearly linked to the text. Therefore, the connection between hypothesis, metabarcoding, morphological analyses, and biological interpretation got lost. The methods are not clearly written and hard to follow. In addition, the discussion is not based enough on the results and is a little shallow. I also suggest an additional statistical test to better compare the genetic identification to morphological data. However, based on the number of samples and only one gene-region the base for this study is not the strongest. It is not weakening the results, but it should be discussed and mentioned for further studies. Especially why some samples failed (not effective storage over years).

Reviewer #2 (Remarks to the Author):

Martinez and colleagues on 'Human access impacts biodiversity of microscopic animals in sandy beaches' use 18S metabarcoding approaches to assess meiobenthic diversity and how tourism could affect such communities. The authors also use morphological characters to assess diversity levels at the different sampling beaches. The manuscript has some interesting ideas but the results and discussion should be improved.

The methods are very well explained and comprehensive, with 11 sampling points, totaling 33 samples and detailed in silico analysis. Molecular ecological analysis comprised phylogeny and community composition combined with classical taxonomy assignments using morphology. Nonetheless, the discussion does not reflect most of the analysis done. This should be improved substantially. The main meiofauna species found should be highlighted, or at least the main phyla found, as well as the reason behind it. The samples look geographically very distinct and so the sampling sites must be ecologically and environmentally different, open areas, how is the composition of the meiobenthic communities throughout the different sites? Table S9 is not good to visualize such

results and these are not mentioned in the discussion in depth. There were many variables measured but mainly depth and trampling by tourists was discussed. Other variables such as grain size known to impact meiofauna communities was not discussed, even if no correlation was found. Could this be due to meiofauna ecology, life strategy or a combination of factors? It would be interesting to have an MDS analysis to better visualize correlations between community composition and environmental variables. The authors also refer to soft-bodied animals, probably platyhelminthes, it would be interesting to know which soft-bodied species were found and why. Were there any differences between samples? What species were found to be overlapping between the two approaches, e.g. venn diagram of morphology vs. HTS. The authors could include 1-2 additional figures including this information.

Pg6|149- 'negative relationship' to 'negative correlation'

Pg6L153- first mention of 'trampling' could be explained i.e., walking, stepping

Pg6L160-'A total of 300 people per day every 10m² means' to 'A total of 300 people per day walking every 10m² means'

Pg7L178- The HTS dataset could be normalized in order to confirm such trampling patterns? since morphology did not reveal these patterns, would be interesting to discuss more in depth as well.

pg7|191-194 the sentence is not very clear

Pg10|268- 'blast assignation' to 'blast taxonomic assignments'

Pg21- MOTUs in the figure and ZOTUs in the legend.

Reviewers' comments:

Reviewer #1 (Remarks to the Author):

The study is analyzing the effect of the human impact of trampling, like a gradient of tourism on Meiofauna diversity at the beach. Analyzed are different beaches at an Island in Italy, with different restrictions of tourists by sampling the three different depth (0cm, 30cm, 2m). The methods are comprehensive including metabarcoding (18S), morphologically based analyses, as well as granulometry. The extensive statistical analyses showed negative indications especially for copepods and Acoela of tourists walking along the waterline. Even if the sampling size is not high and samples were not in optimal condition, the study shows that the metabarcoding approach is a valid and faster method for biodiversity detection of sandy beaches and can be included in the further environmental and touristic decision-making process. Overall the study is convincing and important for that field. It is given a clear question, important for politics and economy, and up to date methods able to answer that. It is a nice example of how we can use Metabarcoding and how to correctly use them for statistical evaluations.

REPLY: we thank the reviewer for the positive assessment. We are glad that the main message we wanted to convey was actually seen by the reviewer.

Within the study, a strong hypothesis is given, nice results and a clear conclusion. The study is valid, based on useful methods and data, but they are not presented in a way that it is obvious. The parts do not seem fitting to each other and the central theme got lost (especially in the introduction and methods). Therefore, I would suggest rewriting parts of the manuscript and being more specific, to better present the scientific context of this study. As I am not a native speaker I can and will not correct language or grammar, however, a style check is always recommended.

REPLY: We changed several parts in the introduction, results, discussion, and methods to provide a more specific goal, and we carefully went through the style of the English language once again. Specific details of the changes are reported in the following answers to the comments.

The text focused a lot about value tourism in the area, however, it seems repeating, suggestive and not on point.

REPLY: We understand the point, and we rewrote some sentences related to the effect of tourists (e.g. line 54-57).

There are clear data for that but in the supplementary and not clearly linked to the text. Therefore, the connection between hypothesis, metabarcoding, morphological analyses, and biological interpretation got lost. The methods are not clearly written and hard to follow. In addition, the discussion is not based enough on the results and is a little shallow. I also suggest an additional statistical test to better compare the genetic identification to morphological data. However, based on the number of samples and only one gene-region the base for this study is not

the strongest. It is not weakening the results, but it should be discussed and mentioned for further studies. Especially why some samples failed (not effective storage over years).

REPLY: Yes, the morphological part was not deeply included in the analyses. The reason is that we have data only for few beaches and few groups. Thus, we report the rationale for not performing any quantitative comparison on line 209-211, and expand this part in the discussion, on line 185-215. Reviewer 2 also suggested improving the comparison between morphology and metabarcoding by making it more quantitative by using venn diagrams. Yet, we cannot perform such analyses due to the high number of undescribed species with no available DNA sequence data, not allowing to match morphology with metabarcoding.

In detail

Abstract

Your abstract is nice, well written and catchy, but please make sure that results are the same as in the actual manuscript.

REPLY: thanks. We now amended the abstract to include more details on the analyses (e.g. granulometry and beach length on line 34, the list of meiofaunal groups on line 29-31, and an explanation on the effect of tourists on line 36).

L 33 “difficulty to access” please be more precise. It is not clear what you mean, sampling, humans or meiofauna.

REPLY: Phrase changed into “fewer tourists present in water away from the beach”, now on line 36.

L33 The indicated negative

REPLY: done.

Introduction

You describe the impact of tourists detailed, but it is most suggestive and you ignore the impact of locals. Therefore, I would recommend to reduce this part and write at least a short sentence about difference of impact by locals.

REPLY: We now make it clear that no local inhabitants exist on the island (on line 83), and only tourists access the beaches of the island, in the methods section on line 253-254.

On the other hand it is described a strong hypothesis, but it is not connected to the introduction or methods at all. Therefore, you could increase that part, and explain with references your hypothesis

REPLY: The introduction was greatly restructured to clarify the hypothesis.

l. 52 you reference mostly the mechanic disturbance of humans. What is with chemical influence, as sunblocker? It is mostly known for corals (Sánchez et al., 2013), but do you think this would have an impact as well?

REPLY: this is a great suggestion. We now include such ideas in the introduction on line 54-57 and in the discussion on line 169-171.

L. 53 the paragraph is connected to the previous one, so please recite all mentioned studies, or avoid the separation of both sentences.

REPLY: references are now reported immediately in the sentence.

l.58 I now there are a lot of different definitions of meiofauna out there, but please try to include a more specific definition with a reference. This will be read by taxonomist, more genetic based people as well as environmental management representatives, it would be helpful if anyone understand clearly the biodiversity level we talking about.

REPLY: reference to the most commonly cited book on meiofauna (Giere, 2019) is now reported to define the term.

L. 60 are you sure to use functional groups in this context?

REPLY: We replaced “functional groups” with “ecological features”, now on line 65.

l. 62 reference 12 , even if a really nice literature need to be included, it is not reflecting the major content of this sentence, please provide additional references.

REPLY: two new more fitting references are now added.

l. 71 undeveloped? Are you sure with that word

REPLY: sentence changed to “least impacted localities with sandy beaches”, now on line 82.

l 71-73. This sentence is confusing on several levels. First, put the next sentence in front, it makes it more smooth for the reader to get the context. Secondly, you ensure that even a small effects of humans could be identified? This is an extremely strong statement, and you give no basics for that. Please remember tourist is not equal to human impact. I believe you that it’s the strongest, so it is fine to focus on that, but you can not prove it is the only one, (see also l.78/79). Its fine to avoid other human impacts in this study, but please avoid such statements.

REPLY: the whole part was rephrased, toning down the effect of low level of tourism.

ll. 74-87 this part is weak and a little suggestive, be more specific or reduce it. AS you have detailed information’s in the methods, think about reducing it.

REPLY: this part has been rephrased clarifying the rationale for the hypothesis.

L 84 This working hypothesis is completely out of context, regarding this introduction. Give references and explanations.

REPLY: the hypothesis is now introduced earlier (on line 50-57) and with ideas that were before present only in the discussion.

l.88 18SrDNA You did not mentioned once genetic analyses within the Introduction once, even it is a major point. To put this in the last sentence is not smooth. Also give references and details about the benefit using 18S metabarcoding of Meiofauna, especially why you choose 18S V1-2.

REPLY: Thanks. We introduce DNA based methods earlier in the introduction, from line 87-90.

Results:

General, the results are good; however I miss the including of morphological detection. This data are valuable and should be included more (its so much work to process them, they deserve better). Could you use the same phyla using morphology for genetics, to see that there is still one or

no effect? Than you probably can show why morphological analyses were not consistent with genetics; or why genetics are better indicators. You clearly see that the phyla that have an effect are not used for morphology. However, you did not discuss that.

REPLY: We agree that the morphological part was not well included in the manuscript. We now include additional information on morphology already from the introduction on line 90 and keep them in the results on line 134-137.

I have little problems to completely understand the results, as I am not sure which data exactly you used quantitative or presence absence, and based on the bioinformatics processing. Did you use the amount of reads or number of ZOTUs? If the methods parts is rewritten, the last inaccuracies should be solved for the reader to understand the results. I suggest you used qualitative data. Currently for metazoans using general diversity we cannot going quantitative, as we have multicellular organism and a high Primer bias. If you used that, I would not reject as I see the effort done with this study. However, I would highly suggest that you include an explanation why, and on which bases you included this data. In addition, it is confusing as morphological data seem on be present absents in the table.

REPLY: Thanks for highlighting these potential misunderstandings. We addressed all the comments for the methods section and we provide some more information on the use of qualitative data only without abundances, already in the results, on line 142.

106 out of curiosity, I suggest that you found more Kinorhyncha with morphology, even if not included in your analyses. Only if you know it.

REPLY: we did not specifically target Kinorhyncha for our morphological analyses, but we indeed found very few animals, in line with the low number of ZOTUs for this group. Yet, to account for the comment, we now mention a potential explanation on the low ZOTU diversity for some groups on line 213-215.

Table 2 on the side is written MOTUs at the text ZOTUs.

REPLY: ZOTUs and not MOTUs is now written as label of the Y axis of figure 2.

143 here is the issue... By Metabarcoding some samples will be always more sequenced in depth than others; Therefore normally the data should be equalized by rarefaction or similar approaches. McMurdie, P. J., & Holmes, S. (2014). Waste not, want not: why rarefying microbiome data is inadmissible. *PLoS computational biology*, 10(4), e1003531. It is the same principle as Microbiotic studies. I am personally not a big fan of rarefaction, so I am not suggesting the reanalysis, but this is the explanation and should be mentioned.

REPLY: thanks for the useful suggestion. We now include the reference and provide an explanation of the rationale of not using rarefaction on line 302-305.

Discussion

The discussion is poorly based on the results. It is shallow.

REPLY: The discussion is now more detailed, with additional ideas also suggested by the reviewers.

152 I wish such a sentence in the introduction and/ or methods, there it was not the clear described. Especially you say mostly, so you acknowledge that there are unseen human influences, which we can ignore here.

REPLY: this idea is now present also in the introduction, together with additional issues related to human presence, on line 54-57 and 169-171.

171 I am not convinced about the message of this sentence. If you do it the right way, metabarcoding on beach samples will work fantastic. I think personally your extraction method is not optimized, which is okay, but not to make generalized statements. You working on bulk samples, you have so much high quality DNA at the beginning. Today we use metabarcoding for eDNA deep sea / ice, air or high degraded DNA with much higher efficiencies. Please rewrite your sentences. And you discussing here you're not perfect samples, however include please also the fact that multigene analyses are more standard today and improving the taxonomic assignment and detection. Only give potential gene-regions (COI is obvious, but ITS or 18S V4/9 could be more fitting for Meiofauna, what do you think about them?)

REPLY: We toned down the message of the sentence, and we include a discussion on using additional markers on line 191-194, 194-196 and 196-201.

173 Important point, but "chemically preserved" seems not fitting. I would suggest something like : non optimal sample processing and storage for DNA analyses.

REPLY: sentence changed

176-181 I am not in agreement with this interpretation. Genetic is faster, less subjective, and mostly cheaper. However, you have no abundances, which is the critical point for applying indexes and evaluating the health of an ecosystem. As the AMBI in marine systems. On the other hand, you did not analyse all groups morphologically, may not the most important groups, which would show the effect. Moreover you give a valuable information, 88 of 196 are new species. A nice fact, which shows the necessity of describing species, but the speeding up by genetics for identification. In this, regard especially when you talk about political and economic application, we missing indicator systems for Metabarcoding. You can cite the gAMBI as future examples (Aylagas, E., Borja, Á., & Rodríguez-Ezpeleta, N. (2014). Environmental status assessment using DNA metabarcoding: towards a genetics based marine biotic index (gAMBI). PloS one, 9(3), e90529.)

REPLY: we tone down the comparison and expand the discussion on the topic on line 203-213, and include the suggested reference.

Methods

The sampling itself is described inefficient. I cannot understand the system of samples per beach. It is shortly explained at line 93, it should stay there but should be explained more specific in the methodology. I am not able to bring your sampling/analyses in consistent with the hypothesis. Please include the reference to your supplementary table S1. Even more important explain in the text what deep, swash, and shallow is, exact definitions please, may a graph or figure? There is no context given, reading it once, it was really confusing. Moreover, you have 11 beaches, with 3 samples per beach I would expect 33 samples.

REPLY: thanks for spotting the potential ambiguities in not understanding the sampling scheme. We tried to address all of the problems, adding some sentences (e.g. line 281, 289-291, 480) and by including figure 1D to explain the sampling scheme.

233. I thought how suggestive, until I realized that the data are included, please link them in the text to the supplementary. I would suggest this table S1 is much too important for supplementary, think about put them in to the MS itself.

REPLY: we provide an explicit reference to table S1 here and in other parts of the text. We would prefer not to add the table in the main body of the manuscript, given that it has data but no results.

242. Two replicates, so you extract twice per sample, leading to 66 samples theoretically. I am confused.

REPLY: sentence rewritten, now on line 289.

246 with not from. Sentence is confusing, rewrite

REPLY: Done.

247 you do not explain or reference MgCL decantation at all. Change it.

REPLY: a reference is now added for the decantation technique.

247, stored in ethanol, for 4 years. Did you cool them? If not, I understand why you lost samples, because there should be more than enough originally. This is not optimal storage for DNA.

REPLY: yes, at -20°C, now on line 287.

248 did you analyses the exact same sampling size for morphologically?

REPLY: yes, the sentence is now clearer, stating that 4 replicates were collected, one of which for morphology, now on line 281 and 331.

248. So you compared to morphological analyses. Nothing written in abstract or introduction. It is a big surprise, and a nice one. However, mention it before, than more people will read it.

REPLY: Thanks: we now added the morphological analyses from the beginning, on line 90.

250. 33 samples.... Can't understand which, please reference again to the sample table.

REPLY: reference to Table S1 now included.

257 explain ZTOUS. It is in the text before, but it is helpful in the methods as well.

REPLY: done.

263 Blast, please be more specific, which algorithm and which reference database. I hope you did not used the 111 Silva database.

REPLY: blast was performed on the whole database from GenBank, not on Silva; the clarification is now on line 307-308.

267 Is that the neighbor-joining tree necessary? Especially based on the high amount of uncultured Eukaryotes or non-Metazoans as *Leptonemella_vicina*. So non-metazoans were note excluded as explained. However, please include detailed information how the neighbor-joining tree were processed.

REPLY: The tree was used to check data reliability, potential problems of singletons with long branches, and to check the taxonomic labels unambiguously for each ZOTU. *Leptonemella* is a name of a sequence from a Blast search through GenBank, and it is a nematode, a metazoan that clustered close to one of our ZOTUs, confirming the ZOTU as a nematode. It is true that several of our sequences did not match identified taxa in GenBank but only unidentified eukaryotes: that is the reason for the iterative phylogenetically informed approach we developed to rescue those sequences assigning them to a phylum, class, or order not only on methods based on genetic distances but also on tree topology. We now explain the use of the NJ tree with more details on line 314-321, providing a stronger rationale for its use.

270-273 How so?

REPLY: we now clearly report the desired taxonomic level on line 320.

273 Which genbank sequences, missing context. And what do you mean by handled sequences.

REPLY: the part on sequences from GenBank was rewritten, also according to the previous comments on the NJ tree.

274 alignments of what? You describe taxonomic assignment already in line 262. You see the whole process is hard to follow as reader. Please rewrite this paragraph.

REPLY: the part on alignment is now moved earlier to line 315.

279 GTR+G model, is there a reference for?

REPLY: we would prefer not to report a reference for GTR model, as we did not report any reference for established methods, such as Yule process, burnin, chains, etc. Yet, if the editorial style of the journal requires it, we can include Lanave et al., 1984, which is the first mention of the model

292 which beaches, and which exact samples? Sadly, not copepods or nematodes would be amazing, but I see the working overload.

REPLY: reference to Table S9 now included to refer to samples.

317; Is the jaccard index used on binomial Data? Please clarify in the text. Especially because no equalization of sequencing data took place, as ex. like rarefaction. I personally don't think it is necessary in this regard, however you should mention why you did not used any rarefaction or similar approaches.

REPLY: Jaccard is now stated to be from binary presence/absence data, on line 366.

319 What do you define as dominant group, please clarify in the text. Based on the PCR bias, and using universal Primers, there is no base for connecting amount of reads to abundance. However, even using dominant group based on the amount of detected taxa can be dangerous in this regard. The chosen 18S region has different variability depending on the presented Phyla. Combined with the lacking references systems, some groups like nematodes or nemertrda will seem to be much less "dominant" as they are in reality. You need to clarify that in your text, but may avoid the synonym dominant for genetic data.

REPLY: we now define dominant groups as the six groups with more ZOTUs and we provide the rationale for it, that is that phylogenetic diversity metrics obtained from trees with few terminals may provide biased and misleading results, on line 367-370.

324: Statistical model: You made my day. You have a nice dataset including your abiotic data, and you handled it, as it deserved to be.

REPLY: Thanks for appreciating out treatment of the data.

Supplementary methods; HTS

9 in water or elution buffer?

REPLY: Elution buffer. Now mentioned in the supplementary methods.

13-16 release 111?, this seems much too old for a publication 2019/2020. The Silva database probably at least doubled until today, especially for meiofauna. Repeat with at least version 132 (most commonly used, and therefore easy to compare) and give detailed information about results, especially regarding the meiofauna of the region. Or let it out. To point out the effectiveness of the chosen primers for your research question is a good point. You can do that better by showing comparable studies using this primers. The given changes should mentioned, but not in that way. You can say it based on an unpublished previous study, it is less suspicious than that. Interested persons can contact you anyway.

REPLY: Yes, this is an old release. The reason is that the improvements on the primers were done when the study was first planned, thus many years ago. We, however, understand that the version used is really old, thus we took out the comment about the coverage of the primer on an old dataset.

26 out of curiosity, why you use 50C and later 62C. Especially without a gradient, this is relatively unusual for 18S barcodes.

REPLY: This bicyclic PCR was also previously used for 18S barcoding as mentioned in the reference that is now added to the text (<https://doi.org/10.1093/femsec/fiw200>). The underlying idea is that the low annealing temperature (50°C) should be relatively unselective followed by a selective step at 62°C. This should increase the abundance of sequence reads belonging to rare taxa in the dataset.

Supplementary table S9 The top of the table is absolute confusing. What is what sample? Bring them in context with S1

REPLY: heading of Table S9 now with additional information to match samples to Table S1.

Reviewer #2 (Remarks to the Author):

Martinez and colleagues on 'Human access impacts biodiversity of microscopic animals in sandy beaches' use 18S metabarcoding approaches to assess meiobenthic diversity and how tourism could affect such communities. The authors also use morphological characters to assess diversity levels at the different sampling beaches. The manuscript has some interesting ideas but the results and discussion should be improved.

REPLY: we changed most of the text to address the comments of both reviewers.

The methods are very well explained and comprehensive, with 11 sampling points, totaling 33 samples and detailed in silico analysis. Molecular ecological analysis comprised phylogeny and community composition combined with classical taxonomy assignments using morphology. Nonetheless, the discussion does not reflect most of the analysis done. This should be improved substantially. The main meiofauna species found should be highlighted, or at least the main phyla found, as well as the reason behind it.

REPLY: Both reviewers suggested to improve the discussion and we now enter in more details for several topics.

The samples look geographically very distinct and so the sampling sites must be ecologically and environmentally different, open areas, how is the composition of the meiobenthic

communities throughout the different sites? Table S9 is not good to visualize such results and these are not mentioned in the discussion in depth.

REPLY: The samples are not very distinct: the maximum distance between beaches is 15km. We also included the effect of differences between beaches in our statistical models. Yet, we acknowledge that the discussion did not mention the issues of confounding factors, which are now included in several parts of the discussion.

There were many variables measured but mainly depth and trampling by tourists was discussed. Other variables such as grain size known to impact meiofauna communities was not discussed, even if no correlation was found. Could this be due to meiofauna ecology, life strategy or a combination of factors?

REPLY: Thanks for the suggestion. We now discuss this issue in more detail, taking advantage of the suggestions of both reviewers.

It would be interesting to have an MDS analysis to better visualize correlations between community composition and environmental variables.

REPLY: We tried to run such analyses, and we found the same effect of granulometry for the first axis of MDS that we found with the PERMANOVA. Yet, PERMANOVA is more powerful, in that it allows to include all the variability, and not separating it axis by axis as in MDS. So, we would prefer to keep the original adonis for PERMANOVA in the manuscript. The MDS plot is also not very informative, and we would prefer not to include it in the manuscript, not even as supplementary material:

We tried also CCA and RDA to test the suggestion of having a plot, but the results are similar to those of the adonis that we already include in the manuscript, and we would prefer to keep the

original structure of the analyses with adonis without a plot to describe the effects of variables on community composition.

The authors also refer to soft-bodied animals, probably platyhelminthes, it would be interesting to know which soft-bodied species were found and why.

REPLY: We now refer to some examples of soft-bodied meiofauna in the discussion (line 189-190), and clearly define them in the methods (line 311-313).

Were there any differences between samples? What species were found to be overlapping between the two approaches, e.g. venn diagram of morphology vs. HTS. The authors could include 1-2 additional figures including this information.

REPLY: Unfortunately, due to the high number of new species (at least 88 out of 196) and to the lack of reference library for most of the other species, we cannot explicitly compare HTS and morphology with quantitative approaches (e.g. venn diagrams). Yet, we understand the problem and we expand this issue in the discussion on line 209-211.

Pg61149- 'negative relationship' to 'negative correlation'

REPLY: done.

Pg6L153- first mention of 'trampling' could be explained i.e., walking, stepping.

REPLY: done. In the same sentence we also add the potential effect of sunscreen creams and bacteria, as suggested by reviewer 1.

Pg6L160-'A total of 300 people per day every 10m2 means' to 'A total of 300 people per day walking every 10m2 means'

REPLY: done.

Pg7L178- The HTS dataset could be normalized in order to confirm such trampling patterns? since morphology did not reveal these patterns, would be interesting to discuss more in depth as well.

REPLY: We now discuss the pattern not only on the basis of trampling but also on other indirect effects (e.g. sunscreen creams or fecal bacteria) on line 169-171. We did not normalize the data, and we explain the reason for that on line 302-305, also according to the comments of the other reviewer.

pg71191-194 the sentence is not very clear

REPLY: sentence rephrased.

Pg10l268- 'blast assignation' to 'blast taxonomic assignments'

REPLY: done.

Pg21- MOTUs in the figure and ZOTUs in the legend.

REPLY: Thanks: we changed the typo.

REVIEWERS' COMMENTS:

Reviewer #1 (Remarks to the Author):

The Manuscript improved strongly. The introduction, methodologies, and results are clear, valid and easy to understand. The discussion improved as well, based better on the data, structured and at least shortly discuss the most important issues.

Besides small remarks, I still have two major comments belonging to the manuscript.

First, the discussion is not valid in its entirety, because it is not always in regard to current literature. Some statements should be toned down or be rewritten.

Secondly, the divergence of morphological and genetic data. Even both reviewers asked for a comparison of both approaches, they were not included. They state it is not possible, but this is simply not true. That there is less overlap is still valuable information. In addition, they argue correctly that that are different methods and therefore give partly different results. The manuscript would actually improve by underlining with this comparison, that the chosen genetic approach is the better approach for this research question. I would highly engage the authors to use the correct argumentation in their discussion-based more on their own data.

Furthermore, I am not a native speaker, but I would engender to do a final style and grammar check before resubmitting. I found some floppy sentences.

Reviewer #2 (Remarks to the Author):

The authors have improved the manuscript substantially according to previous comments focusing on statistical analysis and re-shaping the discussion section. It reads well, especially the introduction and discussion and it is now more focused. It is a pity not see the venn diagram as an overall community composition perspective of both approaches (metabarcoding vs morphology) to be used as future biomonitoring tools, not so much in a comparison perspective but mainly as complementary tools. All techniques have their flaws and despite the authors suggesting poor reference databases to assign all ZOTUs taxonomy, there are other limitations (as primer bias, PCR ect.) as there is for morphology (insuficient taxonomy expertise, different taxonomic assessments). I don't think this is crucial to reiterate the study focus in the field of conservation and ecological assessment using NGS tools and the importance to preserve other natural areas impacted so heavily by human pressure.

Rebuttal letter

Reviewer #1 (Remarks to the Author):

The Manuscript improved strongly. The introduction, methodologies, and results are clear, valid and easy to understand. The discussion improved as well, based better on the data, structured and at least shortly discuss the most important issues.

REPLY: thanks for the positive assessment. We are glad that the reviewer appreciated what we did to improve the manuscript.

Besides small remarks, I still have two major comments belonging to the manuscript.

First, the discussion is not valid in its entirety, because it is not always in regard to current literature.

Some statements should be toned down or be rewritten.

REPLY: We changed some sentences following the detailed suggestions.

Secondly, the divergence of morphological and genetic data. Even both reviewers asked for a comparison of both approaches, they were not included. They state it is not possible, but this is simply not true. That there is less overlap is still valuable information. In addition, they argue correctly that that are different methods and therefore give partly different results. The manuscript would actually improve by underlining with this comparison, that the chosen genetic approach is the better approach for this research question. I would highly engage the authors to use the correct argumentation in their discussion-based more on their own data.

REPLY: the Venn diagrams are now included in figure 3.

Furthermore, I am not a native speaker, but I would engender to do a final style and grammar check before resubmitting. I found some floppy sentences.

REPLY: Thanks for spotting typos and for the grammar suggestions.

In detail

You wrote: Reviewer 2 also suggested improving the comparison between morphology and metabarcoding by making it quantitative by using ven diagrams. Yet, we cannot perform such analyses due to the high number of undescribed species with no available DNA sequence data, not allowing to match morphology with metabarcoding. I am sorry but there is not an acceptable answer. That morphology and metabarcoding do not match well in environmental samples is a well-known fact, which was discussed in several papers over the years. I would recommend reading and may include the following paper: Cowart, D. a., Pinheiro, M., Mouchel, O., Maguer, M., Grall, J., Miné, J., et al. (2015). Metabarcoding is

powerful yet still blind: A comparative analysis of morphological and molecular surveys of seagrass communities. PLoS One, 10, 1–26.

Therefore, to not present such a graph, or any type of analysis, to avoid the wrong interpretation of readers, is simply the wrong way of thinking. It is not weakening your message. 18S V1-V2 is perfect for meiofauna but hardly working for species level, so it is easy to explain while the Venn diagram on the genus level showed hardly an overlap. Thus, you should show the diagram on the genus but also on a higher level, as a family level. Even if a species is not in the database, you should be able to identify the sequences as family or phylum. Therefore, you can easily underline the fact that with Metabarcoding you, in fact, find higher biodiversity. You mentioned these very important facts in the discussion; underline this please with presenting the data in any graphic way. Having different results with different methodologies, not meaning that one is right and the other wrong. You simply underline your message, that you need morphological analyses for describing new species, but that metabarcoding in your regard works more efficiently to detect biodiversity.

REPLY: We include the Venn diagrams as figure 3 in the new version of the manuscript.

A note: Please do not forget, all not assigned DNA sequences can be both, undescribed species as well as a lot of sequencing trash.

REPLY: well, all sequences could be assigned to some groups higher than family level, so they may not be sequencing trash. We do not add any sentence in the manuscript regarding this issue.

Abstract well done, only one note to improve

L.36 water depth (xxxx) please give an indication of which depth you measured, like: up to 2m

REPLY: Thanks, done.

Introduction

Nicely improved, it is clear, sharp and well written. A pleasure to read. Only small comments for the end.

REPLY: Thanks. All the following suggestions have been included in the new version.

l.87 “as a whole as a metric for biological diversity”. I am not a native speaker but sound a little awkward. I suggest this “To apply meiofauna as a metric for biological diversity...”

l.88 Please be specific 18S V1-V2, you did not analyze the whole 18S, which is currently not possible with metabarcoding However, it is an important fact which part you used.

l.90 “with an attempt” don’t be too harsh to yourself! One or more of your co-authors invested a lot of work into this, and I believe it is a well-done job. Even if

you focus more on metabarcoding, the morphological aspect of your study deserves more than a half-sentence, where you particular offending it to the same time. It is one thing to engage metabarcoding for detecting biodiversity, but we still need taxonomists to describe species. I think this is your message too, so please feel free to include this more obvious (may not in the introduction).

l. 91 This whole sentence makes literally no sense. "working hypothesis to identify the effect"? You can write: "our aim was to identify", or "our hypothesis was a negative effect...". The same with "is that any effect will be detected more intensely"?... Maybe two shorter sentences would be easier to understand.

l.92 delete the "figure 1C "while explaining your "hypothesis" we do not need to see a picture of a beach. The same for figure 1D inline 96.

l. 95-96 Please delete the i.e. in this whole paragraph, it is theoretically correct, but practically a little confusing and unnecessary in this context.

Methods

I am old school I read them first.

REPLY: The figure caption now clearly names the beaches. All other minor issues have been addressed.

l. 264 the named beaches are not clearly indicated in the graph. However, if you mentioned it, it would be good to know, can you include that into the graph 1B, please. Alternatively, give the exact sampling site in the text, and number all sites in graph l.

268 you linking to figure 1D and E. This is out of context, one is a photo the other results,

L.272 "beaches is on the number" cut out the "on"

L 273 the sentence "The daily ..." is hard to read an includes several grammar issues. As an example: It is tourists and a few.

L 286. Thanks for giving the citation, but may put it to the decantation, not to the "63um" and be more specific. The resource is a book, I am not sure with the guidelines of this journal but it would be more helpful having at least the chapter.

289-292 change both sentences, the second should be first. You included now the correct numbers, but now out of context. You don't explain that you used three samples per site. And starting with 66 the paragraph leaves me only with question marks. Link the appendix (SI Appendix) again. I should be able to follow the basic process even if I did not read the appendix or the results.

328 GTR model its okay not to reference, but at least write down the full name, and not only the shortening

Results

The results are clear understandable written.

REPLY: Thanks.

Discussion

The discussion is mostly valid. However, I (personally) have the impression that the interpretation should be forced into the direction which is seemingly not completely fitting.

I see the point in this study not that it shows the effect on tourism at beaches really well, because it is local, the sampling and sequencing not that great, and the impact is only in the squash zone. What you showed is that you applied new methodologies showing better biodiversity than the current standard and really filling a need to analyze possible impact. At a place, which shows to be the perfect place to further test evaluate and develop it. You mostly write correctly that tourism is a “maybe” effect at Meiofauna. It is a wonderful first study in this regard, so don’t go too far with your discussion. Also, make sure it seems not too aggressive.

REPLY: We followed all the suggestions, especially in the light of toning down excessive conclusions and of clarifying the two main messages and using them in the same order throughout the text.

192. “now a reality” I love your enthusiasm but I am sorry, could you please rewrite this? It is still a scientific journal. It could be offending depending on the background. More something like it is getting common approach science for several years. Or so.... There is a lot of literature about meiofaunal metabarcoding, less for the beach but more for meiobenthos. In this regard, some colleges published this recently: Rossel, S., Khodami, S., & Martinez Arbizu, P. (2019). Comparison of rapid biodiversity assessment of meiobenthos using MALDI-TOF MS and Metabarcoding. *Frontiers in Marine Science*, 6, 659. This could be fitting in your discussion quite well.

REPLY: sentence rephrased.

193. To use metabarcoding for environmental monitoring is not new, people work on it for over 10 years. Most currently used the idea for eDNA. But we mostly use this term with quotes, as we get no abundance data, no current reliable indexes and we cannot confirm in total that what we find, is actually living there. As you found non-meiofaunal in your preselected samples, how you explain that? So please rewrite a little.

As I understand you show an interesting new application and new political branch for this already worldwide idea. I would love you to emphasize this within this context.

REPLY: sentence rephrased stating that it is “now a common approach in biodiversity studies”.

197 “making meiofauna a useful candidate to develop ecological indexes of the biological integrity of coastal areas 30.” Delete this part, you going too far, especially using this study as a reference. In the following, you are going to this in better-detailed argumentation.

REPLY: sentence deleted.

209 -213 Here I the part which I addressed before. Of course, you can compare it, you simply did not do it. No overlay is still a result, you even state that it provides different results. But still, you can do it.

REPLY: Venn diagrams included as figure 3.

226-228 This is a really hard statement, and it breaks down a really complex issue in a not comfortable way. I mean we all wish it would be that way, but in fact, there are reasons it is not.

REPLY: sentence removed.

I am only confused about whether you are sure what your message is. Your discussion seems to be more focused on the environmental protection effort and seems more to address such associated people, than on the method and the possibilities of this application itself. On the other hand, this political-economical issue is broke down in a really offending way for such readers. Still how valid can you discuss this really on that data?

REPLY: We now divide more clearly the discussion on methodological issues and on political-economical issues, making the latter shorter.

237. Someone could argue, that you only found an impact at the squash zone, so may you can give an additional specific argument in this regard. How this could affect the rest of the ecosystem?

REPLY: we now include this caveat in the sentence.

Conclusions

First of all, point two should be point one, for logic sake. Remind my comments above. Again please rewrite it as a "future possible" methodology. Make sure you are closing with your hypothesis from the introduction.

REPLY: order rearranged.

Actually, stakeholders and industry they want to use Metabarcoding most urgently, but what we currently see is not a clear impact, mostly only OTU tables. There are still limitations, which still need to overcome. So write, as a possible future application would be more fair, reliable and less aggressive.

REPLY: we now mention it as a "reliable future possible methodology".

Reviewer #2 (Remarks to the Author):

The authors have improved the manuscript substantially according to previous comments focusing on statistical analysis and re-shaping the discussion section. It reads well, especially the introduction and discussion and it is now more

focused. It is a pity not see the venn diagram as an overall community composition perspective of both approaches (metabarcoding vs morphology) to be used as future biomonitoring tools, not so much in a comparison perspective but mainly as complementary tools. All techniques have their flaws and despite the authors suggesting poor reference databases to assign all ZOTUs taxonomy, there are other limitations (as primer bias, PCR ect.) as there is for morphology (insufficient taxonomy expertise, different taxonomic assessments). I don't think this is crucial to reiterate the study focus in the field of conservation and ecological assessment using NGS tools and the importance to preserve other natural areas impacted so heavily by human pressure.

REPLY: we now include the Venn diagrams.